# A lipid bound actin meshwork organizes liquid phase separation in model membranes

Alf Honigmann[1]*, Sina Sadeghi[2], Jan Keller[1], Stefan W Hell[1], Christian Eggeling[1,3], Richard Vink[2]*

[1]Department of NanoBiophotonics, Max-Planck-Institute for Biophysical Chemistry, Göttingen, Germany; [2]Institute of Theoretical Physics, Georg-August-Universität Göttingen, Göttingen, Germany; [3]Weatherall Institute of Molecular Medicine, University of Oxford, Oxford, United Kingdom

**Abstract** The eukaryotic cell membrane is connected to a dense actin rich cortex. We present FCS and STED experiments showing that dense membrane bound actin networks have severe influence on lipid phase separation. A minimal actin cortex was bound to a supported lipid bilayer via biotinylated lipid streptavidin complexes (pinning sites). In general, actin binding to ternary membranes prevented macroscopic liquid-ordered and liquid-disordered domain formation, even at low temperature. Instead, depending on the type of pinning lipid, an actin correlated multi-domain pattern was observed. FCS measurements revealed hindered diffusion of lipids in the presence of an actin network. To explain our experimental findings, a new simulation model is proposed, in which the membrane composition, the membrane curvature, and the actin pinning sites are all coupled. Our results reveal a mechanism how cells may prevent macroscopic demixing of their membrane components, while at the same time regulate the local membrane composition.

**\*For correspondence:**
ahonigm@gwdg.de (AH); vink@theorie.physik.uni-goettingen.de (RV)

**Competing interests:** The authors declare that no competing interests exist.

**Reviewing editor**: Randy Schekman, Howard Hughes Medical Institute, University of California, Berkeley, United States

## Introduction

The lateral heterogeneity of lipids and proteins in the plasma membrane of eukaryotic cells is an important feature for regulating biological function. The most prominent concept for membrane organization, the lipid raft theory, relates lipid phase separation (driven by interactions between cholesterol, sphingolipids, and saturated phospholipids) to membrane protein partitioning and regulation (*Simons and Ikonen, 1997*; *Simons et al., 2011*). Consequently, understanding lipid phase separation in membranes is a topic of extreme interest. A convenient starting point is to envision the membrane as a two-dimensional (2D) fluid environment through which the various membrane components freely diffuse. This simple picture successfully captures ternary model membranes containing two phospholipid species and cholesterol. At low temperature, these systems macroscopically phase separate into liquid-ordered (Lo) and liquid-disordered (Ld) domains (*Veatch and Keller, 2003*) and the nature of the transition is consistent with that of a 2D fluid (*Honerkamp-Smith et al., 2008*, *2009*). Similar behavior was observed in plasma membrane-derived vesicles (*Baumgart et al., 2007*; *Sezgin et al., 2012*).

Despite these successes for model membranes, there is growing consensus that this simple picture needs to be refined for the plasma membrane. For example, a remaining puzzle is that the Lo/Ld domains observed in model membranes grow macroscopic in size (micrometers), whereas lipid domains in plasma membrane are postulated to be tiny (nanometers) (*Lenne and Nicolas, 2009*). Additionally, the temperature $T_c$ below which Lo/Ld domains start to form in plasma membrane derived vesicles is distinctly below T = 37°C (*Sezgin et al., 2012*), and so its relevance at physiological

**eLife digest** All cells are surrounded by a lipid membrane that protects the cell, controls the movement of molecules into and out of the cell, and passes messages about environmental conditions to the cell. This membrane is made of two layers of molecules called lipids, with various proteins embedded in it. There are many different types of lipid molecules that together help to keep the membrane flexible. Moreover, lipid molecules of particular types can also come together to form 'rafts' that help the membrane to carry out its various roles.

Given the complexity of the cell membrane, cell biologists often use simpler model membranes and computer simulations to explore how the different types of lipid molecules are organized within the membrane. According to the 'picket fence' model the cell membrane is divided into small compartments as a result of its interaction with the dense network of actin fibers that acts as a skeleton inside the cell.

Recent computer simulations have predicted that these interactions can influence the distribution of lipids and proteins within the membrane. In particular, they can prevent the drastic re-arrangement of lipids into regions of high and low viscosity at low temperature. This temperature dependent re-arrangement of the membrane is known as lipid phase separation.

Honigmann et al. have now used computer simulations and two advanced techniques—super-resolution optical STED microscopy and fluorescence correlation spectroscopy—to explore the properties of a model membrane in the presence of a dense network of actin fibers in fine detail. The results show that, in agreement with the simulation predictions of the 'picket fence' model, the actin fibers bound to the membrane prevent lipid phase separation happening at low temperatures. Moreover, the actin fibers also help to organize the distribution of lipids and proteins within the membrane at physiological temperatures. Honigmann et al. also suggest that the actin fibers cause the membrane to curve in a way that can reinforce the influence of the 'picket fence'.

The results show that the 'raft' and 'picket fence' models are connected, and that a cell can control the properties of its membrane by controlling the interactions between the membrane and the actin fibers that make up the skeleton of the cell.

temperature requires further justification. In the case of a free-standing membrane (i.e., in the absence of an actin cortex), experiments have shown that lipid domains at temperatures above $T_c$ can be induced by crosslinking low abundant membrane constituents (*Hammond et al., 2005*; *Lingwood et al., 2008*). Furthermore, there are numerous theoretical proposals of how a finite domain size above $T_c$ might be accounted for: vicinity of a critical point (*Honerkamp-Smith et al., 2009*), hybrid lipids (*Palmieri and Safran, 2013*), coupling between composition and membrane curvature (*Schick, 2012*), electrostatic forces (*Liu et al., 2005*).

In addition to this, the cortical cytoskeleton has also been identified as a key player affecting membrane domain formation (*Kusumi et al., 2005*). The latter is a dense fiber network of actin and spectrin on the cytoplasmic side of the eukaryotic plasma membrane. This network is connected to the membrane via pinning sites, such as lipid-binding proteins, transmembrane proteins, or membrane-attached proteins (*Janmey, 1998*; *Mangeat et al., 1999*; *Janmey and Lindberg, 2004*; *Saarikangas et al., 2010*). This has led to the hypothesis of the membrane being laterally compartmentalized: the pinning sites structure the membrane into small compartments whose perimeters are defined by the underlying actin network (the so-called 'picket-fence' model). This picket-fence network then acts as a barrier to diffusion, which elegantly accounts for confined diffusion of lipids and proteins observed in a single molecule tracking experiments (*Kusumi et al., 2005*). In a recent series of simulations, it was subsequently shown that a picket-fence network also acts as a barrier to macroscopic phase separation of lipids (*Ehrig et al., 2011*; *Fischer and Vink, 2011*; *Machta et al., 2011*). Instead, a stable mosaic of Lo and Ld domains is predicted, with a domain structure that strongly correlates to the actin fibers. Moreover, this mosaic structure already appears at physiological temperatures. These simulation findings are promising in view of the lipid raft hypothesis, since rafts are postulated to be small, as opposed to macroscopic.

In this paper, we present the first experimental confirmation of these simulation results. To this end, we use an in vitro model system consisting of a supported lipid bilayer bound to an actin network.

Complementing previous studies (*Liu and Fletcher, 2006*; *Subramaniam et al., 2012*; *Heinemann et al., 2013*; *Vogel et al., 2013*), our system enables direct observation of Lo/Ld domain formation in the presence of a lipid bound actin network using superresolution STED microscopy (*Hell and Wichmann, 1994*; *Hell, 2007*) and fluorescence correlation spectroscopy (FCS) (*Magde et al., 1972*; *Kim et al., 2007*). Based on our results, we propose an extension of the picket-fence model by including a coupling of the local membrane curvature to the membrane composition. Computer simulations of this extended model show that the pinning effect of the actin network is dramatically enhanced by such a coupling. These results imply that even a low density of pinning sites can induce significant structuring of lipids and proteins in the plasma membrane.

## Results

### Experimental results

#### Domain formation in membranes without actin

Our model system is a lipid mixture of unsaturated DOPC, saturated DPPC, and cholesterol. Such ternary mixtures are routinely used to approximate the complex lipid composition of the eukaryotic plasma membrane (*Dimova et al., 2006*). The phase diagram for this system in the absence of actin is well known (*Veatch and Keller, 2003*). For a large range of compositions this system reveals macroscopic Lo/Ld phase coexistence below the transition temperature $T_c$, while above $T_c$ they are homogenously mixed. The size, shape, and the exact phase transition temperature of Lo/Ld domains depend on the specific model membrane system used. Domains grow largest in unsupported membranes (*Honigmann et al., 2010*), they are smaller in Mica supported membranes (*Jensen et al., 2007*) and are below the diffraction limit on glass-supported membranes (*Honigmann et al., 2013*). Depending on the composition, the transition to the coexistence region can be first-order, or continuous passing through a critical point. In the present work, we choose a composition ratio of (DOPC:DPPC:cholesterol) = (35:35:30) mol%, including 1 mol% of a biotinylated lipid (DOPE-biotin) to eventually connect the membrane to an actin network. To facilitate high-resolution microscopy flat, single membranes of that mixture were prepared on a Mica support. After membrane preparation the sample was heated to T = 37°C for 15 min to equilibrate the bilayer in the mixed phase. At this stage, the connector protein streptavidin was bound to the biotinylated lipid (but not yet to actin). After this preparation the membrane was slowly cooled and the distribution of the red fluorescent Ld-marker (DPPE-KK114) was determined by superresolution STED imaging. Visual inspection of images revealed that, upon cooling, there is a well-defined temperature at which Lo/Ld domains became visible (*Figure 1A*). This temperature is around $T_c \approx 28°C$ and thus marks the transition between the mixed one-phase state at high temperature, and the low-temperature two-phase coexistence state. This phase transition temperature is close but slightly decreased compared to the reported value for lipid vesicles made of the same mixture ($T_c \approx 30°C$ [*Veatch et al., 2004*]). At $T_c$, domains were constantly forming and re-forming, implying that the line tension is small, and so the transition is close to critical. The transition from a one-phase to a Lo/Ld separated two-phase membrane can be analyzed more quantitatively via the cumulant $U_1$ of the images (see 'Materials and methods' for definition). In the one-phase region, $U_1 = \pi/2$, whereas for a two-phase system $U_1 = 1$ (*Binder, 1981*; *Fischer and Vink, 2011*). As shown in *Figure 1C*, $U_1$ markedly dropped toward unity below $T_c$. Once the membrane enters the two-phase region, domains were seen to coarsen within a couple of minutes. The typical domain size $R$ was obtained from the radial distribution function $g(r)$ of the images. As shown in *Figure 1D*, $R$ increased rapidly at $T_c$, from <75 nm to >200 nm, and continued to grow into the near micrometer range as the temperature was lowered further. These findings are consistent with atomic force microscopy experiments (*Connell et al., 2013*), where the transition was shown to be continuous as well. The observation that the domains do not fully coalesce at low temperature (as opposed to free-standing membranes) can be attributed to the interaction of the membrane with the mica support (*Jensen et al., 2007*). Our data do not allow the nature of the transition to be unambiguously identified. We emphasize, however, that this does not affect the overall conclusions of our work. All that we require is that the system without actin reveals two-phase coexistence at low temperatures.

#### Domain formation in membranes with actin

Having characterized the phase behavior of the membrane without actin, we heated the same membrane back to the mixed state at T = 37°C to bind actin fibers via phalloidin–biotin to the

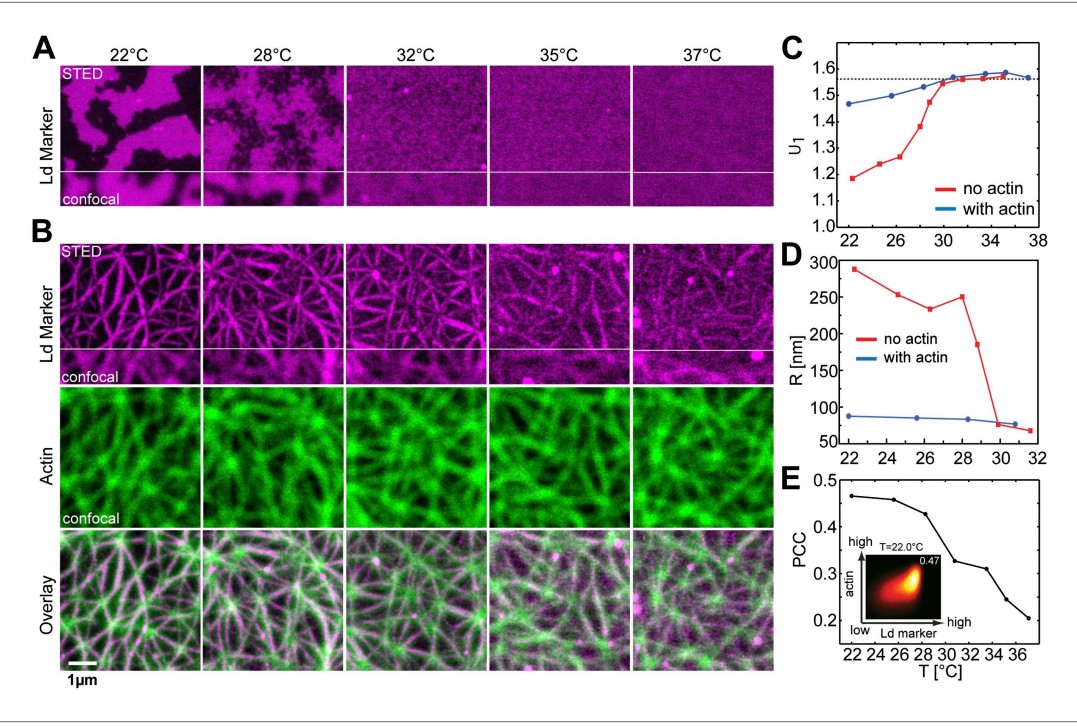

**Figure 1**. Binding of an actin network to ternary membranes dramatically affects lipid domain structure. (**A**) In the absence of actin, our Mica supported model membrane revealed a phase transition at $T_c \approx 28°C$, below which macroscopic Lo/Ld phase separation was observed. The membrane was stained with the Ld-marker DPPE-KK114 (magenta), and imaged by STED-microscopy with a lateral resolution of 70 nm. (**B**) The same membrane as in (**A**) but now in the presence of an actin network (green) bound to the membrane via a streptavidin linker. The Ld domains were strongly correlated to the actin network, as can be seen in the overlay, resulting in a meshwork like structure of Ld channels. This structure was stable even at temperatures above $T_c$ (**C**) Variation of the cumulant $U_1$ of the images with temperature $T$ in the absence of actin (red) and presence (blue). In the absence of actin, there is a transition toward a two-phase coexistence region at low temperature, as manifested by a pronounced drop of $U_1$ toward unity. The transition occurs at $T_c \approx 28°C$. In the presence of actin, no such transition is detected. (**D**) Typical domain size $R$ as a function of $T$. In the absence of actin, there was pronounced domain coarsening below $T_c$. In the presence of actin, $R$ was essentially temperature independent, and restricted to at most 90 nm. Note that $R$ could not be measured for temperatures above T = 32°C as the contrast of the domains became too low. (**E**) Pearson correlation coefficient (PCC) between Ld domains and actin vs temperature. This coefficient measures the degree of correlation between Ld domains and the actin fibers. The correlation is largest at low temperature, but it persists at high temperature also, including the physiological temperature T = 37°C. Inset shows the graphical representation of the correlation between high intensities in the actin channel with high intensities in the lipid channel (ld phase). For another example of phase reorganization by actin binding see *Figure 1—figure supplement 1*. For actin meshwork binding to single component DOPC membranes see *Figure 1—figure supplement 2*.

The following figure supplements are available for figure 1:

**Figure supplement 1**. Phase organization by the actin-network.

**Figure supplement 2**. Actin-network density and lipid diffusion dependence on the concentration of biotinylated lipids. The data refer to a single lipid species (DOPC) supported bilayer at T = 22°C.

**Figure supplement 3**. Molecular sketch of the assembled components in our model system.

**Figure supplement 4**. Optical setup for STED imaging and scanning FCS with pulsed excitation and pulsed STED laser and according beam paths (excitation orange/blue and STED dark red).

streptavidin–biotin–lipid complexes (a molecular sketch is provided in *Figure 1—figure supplement 3*). The resulting binding sites strongly attract the Ld phase with a partitioning value of Lo% = 10 for DOPE-biotin (*Figure 3—figure supplement 1*). The actin fibers were stained with a green fluorescent phalloidin. After the actin meshwork was bound to the membrane, the temperature was again decreased, and the distributions of the Ld marker DPPE-KK114 and the actin were imaged simultaneously using (superresolution) STED and (diffraction limited) confocal microscopy, respectively. Visual inspection of images (*Figure 1B*) revealed that, with actin, domains already appeared at T ≈ 37°C. The spatial domain structure, however, was very different compared to the actin-free case. We observed a partitioning of the membrane into compartments of Lo-enriched domains, separated by 'channels' of Ld-enriched domains. These channels were clearly correlated to the actin fibers (middle row), as shown in the overlay (lower row). This correlation was analyzed quantitatively via the Pearson measure (PCC), which is an estimate of the colocalization between the Ld domains and the actin fibers (see 'Materials and methods' for definition). In general, a large value PCC>0 indicates a pronounced overlay of structures, PCC = 0 indicates randomly distributed objects, whereas PCC<0 represents inverted features (i.e., anti-correlations). We observed that PCC is largest at low temperature, implying that the correlation between Ld domains and actin is strongest there (*Figure 1E*). However, PCC remains finite at high temperatures also, with a significant correlation up to our highest accessible temperatures of T = 37°C, which is a stunning 9°C above $T_c$ of the actin-free membrane. For T > 28°C, the decrease of PCC is approximately linear. This decrease was caused by an increasingly even partitioning of the Ld-marker, and not by a redistribution of Ld domains away from the actin fibers (as follows from *Figure 1B*, where correlated domains remain clearly visible at T = 37°C). In contrast to the actin-free case, our data suggest that there is no phase transition associated with the formation of the domain structure: Neither the cumulant nor the domain size revealed any pronounced temperature dependence (*Figure 1C,D*). Also the variation of PCC with temperature is entirely smooth, and does not indicate a transition either. Furthermore, in the presence of actin, the average domain size was significantly smaller, R ≈ 90 nm at most, and no coarsening was observed. Our conclusion is that the phase transition of the actin-free membrane is effectively eliminated by the actin network, in line with theoretical expectations (*Fischer and Vink, 2011*; *Machta et al., 2011*).

As control experiment, we also considered a single component bilayer (DOPC) bound to actin. In this case, no domains were induced by the actin (*Figure 1—figure supplement 2*). Additionally, we excluded the possibility that the actin-correlated phases were induced by the green-fluorescent phalloidin. To this end, we stained the membrane with the Ld-marker (red) and the Lo marker (green) but not the actin itself. The result was a comparable structure as observed in *Figure 1B*, see *Figure 1—figure supplement 1*.

## The lateral diffusion of lipids is restricted by actin-organized domains

To determine the lateral movement of lipids in the membrane in relation to the actin fibers, we applied scanning FCS (*Digman et al., 2005*; *Ries and Schwille, 2006*; *Digman and Gratton, 2009*). We used the same system as described above, but now with a lower density of actin such that single fibers could be resolved by confocal microscopy. The temperature was set to T = 32°C, that is slightly above $T_c$ of the actin-free membrane. As a control, we started with actin fibers on a single component DOPC membrane including 1 mol% DOPE-biotin (compare *Figure 1—figure supplement 2B*). A small circle (diameter d = 500 nm) was scanned over a single actin fiber, as depicted in *Figure 2E*. For each position on the circle, the diffusion of the Ld-marker was determined by standard FCS analysis (*Kim et al., 2007*). The upper panel in *Figure 2A* shows the intensity of the Ld-marker (red) and of the actin marker (green) along the scan circle for the single component membrane (A). The intensity maximum of the green channel indicates the position of the actin fiber (corresponding to 0 and 180°), the intensity of the red channel shows the corresponding distribution of the Ld marker. In the single component membrane, the Ld marker was homogenously distributed. The lower panels of *Figure 2A* shows the decay of the autocorrelation function of the Ld marker measured along the scan circle, with the color representing the amplitude of the correlation. For the single component membrane, the autocorrelation analysis revealed no significant differences in mobility between areas with actin and without. Next, we used the same data to determine the diffusion of lipids across the scanning circle by calculating the pair-correlation function of 'opposing' pixels on the scanning circle, that is pixel pairs that are separated by a rotation of 180° (*Cardarelli et al., 2012*). This correlation measures how long the probe on average needs to diffuse across the circle and is therefore more sensitive to detect diffusion barriers than the autocorrelation analysis of a single excitation spot. The resulting pair-correlation is shown in

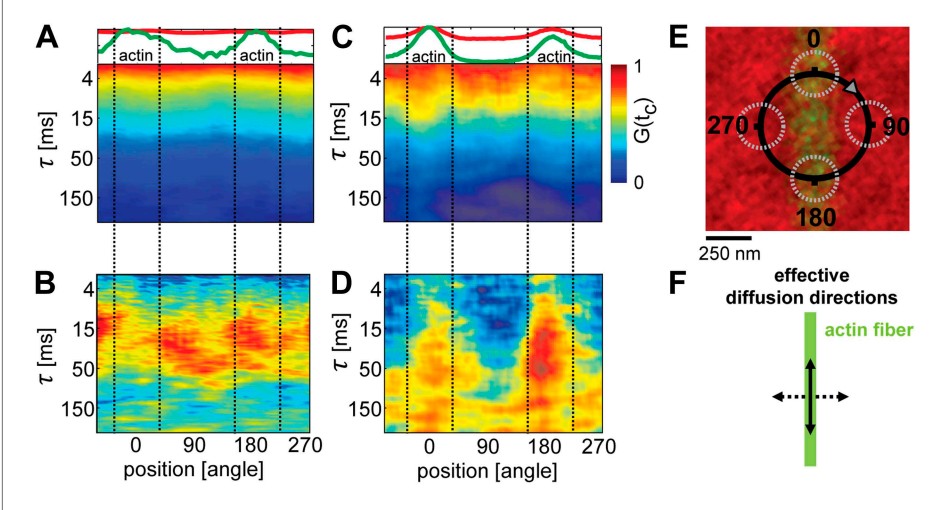

**Figure 2**. Lipid diffusion was restricted by actin-organized domains. (**A**) Scanning-FCS (mobility) analysis of the Ld marker DPPE-KK114 in a single component DOPC membrane in the presence of a low density actin meshwork. The upper panel shows the intensity of the green channel (actin) and red channel (Ld marker), indicating the position of the actin fiber on the scan orbit (perpendicular lines). The lower panel depicts the auto-correlation decay for each pixel along the circular scan orbit with a diameter of d = 500 nm. The normalized auto-correlation amplitude is represented by the color. The mean transit time through the excitation spot (mobility) can be estimated by the transition from yellow to green. As expected the mobility along the scan orbit was homogenous. (**B**) Pair-correlation analysis of the same data as in **A** for opposing pixels on the scan orbit. The maxima in the pair-correlation represent the average time the probes need to move across the scan orbit. No significant directional dependence of the mobility was observed in case of simple one component membranes. (**C**) Same as in (**A**) but for ternary membranes (same composition as in *Figure 1*). (**D**) While the auto-correlation analysis seemed to be homogenous a distinct directional dependence of diffusion was revealed by the pair-correlation. In the direction along the actin fiber, a distinct correlation peak is visible with a maximum at τ ≈ 40 ms. In the perpendicular direction, the correlation amplitude is reduced, which indicates a diffusion barrier along this axis. (**E**) Representation of the scanning orbit over a single actin fiber bound to the membrane. The numbers represent the angles of the orbit (**F**). Qualitative representation summarizing the results of the pair-correlation analysis with pronounced diffusion along (drawn arrow) and restricted diffusion perpendicular to the actin fiber (dashed arrow).

*Figure 2B*. For the single component membrane, the amplitude and correlation time revealed no clear directional dependence. The average correlation time across the circle was τ ≈ 30 ms, indicating that the pinning of the actin filament imposed no significant diffusion barrier for lipids. These results are in agreement with point FCS measurements on single component membranes in the presence of increasing actin densities, which are reported in *Figure 1—figure supplement 2E*. Also here, the lateral diffusion was remarkably insensitive to the presence of actin.

In contrast to the single component membrane, a pronounced directional dependence of lipid diffusion was observed in the ternary system. The intensity distribution of the Ld-marker showed a slight increase at the position of the actin fiber (upper panel *Figure 2C*), indicating a stabilized Ld phase along the actin fiber. While the autocorrelation analysis along the scan trajectory indicated no clear heterogeneities (lower panel *Figure 2C*), the pair-correlation function revealed strong peaks at 0 and 180° (i.e., along the actin filament) at delay times τ ≈ 30–50 ms (*Figure 2D*). In the direction perpendicular to the actin fiber, a much weaker amplitude was observed, with the peak shifted to longer times τ > 50ms. This indicates that the actin stabilized Ld domains favored the diffusion of the Ld marker within the domain (and thus along the actin fiber), while restricting diffusion across domain boundaries. These findings strikingly confirm the simulation prediction of *Machta et al. (2011)*, where lipid domains stabilized by actin pinning were also found to 'compartmentalize' lipid diffusion.

## Influence of the type of lipid-pinning site on domain structure
We next consider how domain formation is affected when different pinning sites are used to bind the membrane to the actin network. To this end, we compared three biotinylated lipids with different

partitioning values: DOPE-biotin (i.e., the same as used in the experiment of *Figure 1*) that partitions strongly into the Ld phase (Lo% = 11 ± 2), DSPE-PEG-biotin that partitions predominantly in the Lo phase (Lo% = 78 ± 7), and DPPE-biotin that partitions almost equally in both phases (but with a small preference toward the Lo phase, Lo% = 59 ± 5). The experimental procedure to determine the Lo% is outlined in *Figure 3—figure supplement 1*. In *Figure 3A*, we show confocal images of the domain structure for the Ld-preferring pinning lipid DOPE-biotin. The figure confirms the previous experiment of *Figure 1B*: We again observe the formation of Ld domains along the actin fibers, as expressed by a positive Pearson coefficient PCC = 0.55 ± 0.02. Pinning of the Lo-preferring lipid DSPE-PEG-biotin induced an 'inverted' domain structure, yielding a negative PCC = −0.37 ± 0.04

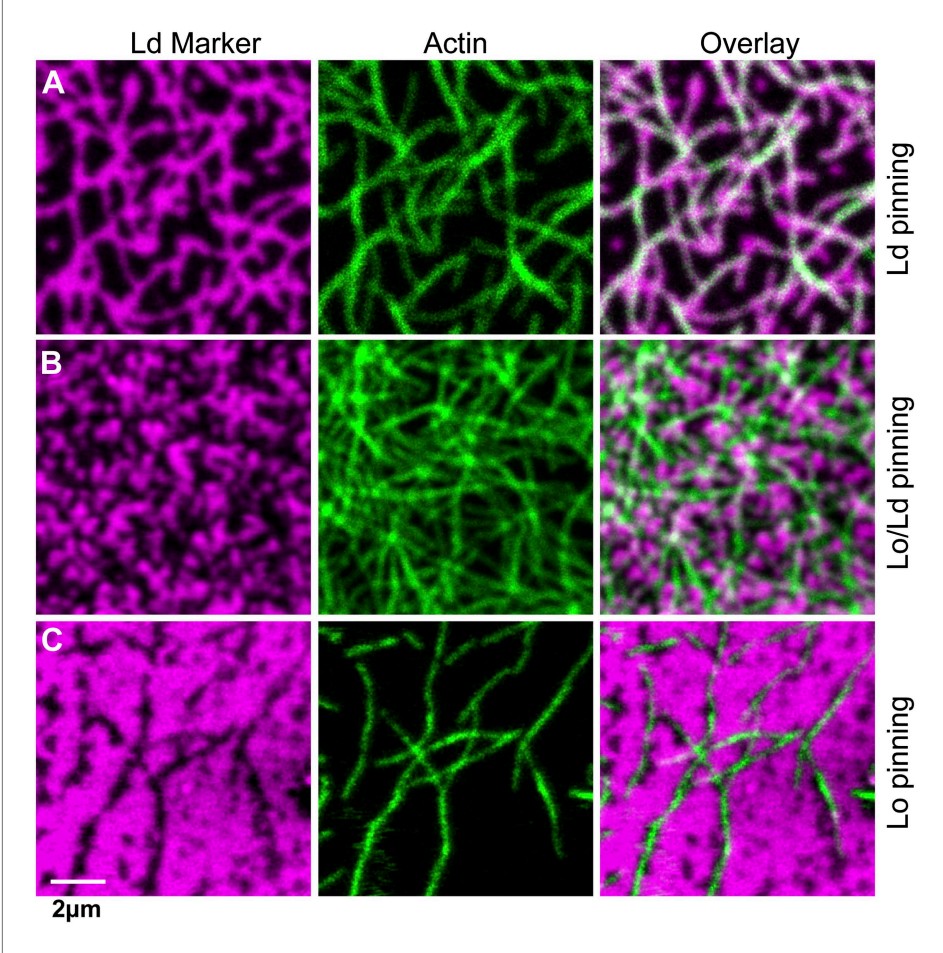

**Figure 3**. The type of pinning site used to bind actin to the membrane strongly affects the domain structure. The Ld phase (left column, magenta) were stained with DSPE-KK114, while actin (middle column, green) was stained with phalloidin-488. The right column shows the overlay of both images. The membrane was imaged by confocal microscopy at T = 19°C, using the same lipid composition as in *Figure 1*. (**A**) Binding of actin to the Ld preferring lipid DOPE-biotin resulted in Ld domains along the actin fibers (as in *Figure 1B*, PCC = 0.55 ± 0.02). (**B**) When actin was bound to DPPE-biotin, the correlation of domains with the actin fibers was significantly reduced, but remained detectable, with a slightly positive Pearson coefficient (PCC = 0.07 ± 0.03). (**C**) Binding of actin to the Lo preferring lipid DSPE-PEG-biotin resulted in correlated Lo domains along the actin fibers, that is the 'inverse' structure of (**A**). In this case, the Pearson coefficient was negative (PCC = −0.37 ± 0.04). For the lipid phase partitioning values of the biotinylated lipids used in this experiment see *Figure 3—figure supplement 1*.

The following figure supplements are available for figure 3:

**Figure supplement 1**. Lipid phase partitioning of biotinylated lipid–streptavidin complexes without actin.

(*Figure 3C*). The binding of actin to DPPE-biotin did not result in a strong localization of lipid domains along the fibers, as manifested by a small 'but positive' PCC = 0.07 ± 0.03 (*Figure 3B*). The result of *Figure 3B* is surprising, since DPPE-biotin slightly prefers the Lo phase, and so a 'negative' PCC was expected. Notwithstanding that several mechanisms could be responsible for this (e.g., streptavidin binding to biotinylated lipids may induce lipid disorder by steric or electrostatic interactions with the membrane), it is interesting that a coupling between membrane composition and curvature also brings about this effect. To illustrate this point, we resort to computer simulations, where important parameters such as pinning density and phase partitioning of pinning sites can be systematically varied.

## Simulation results

We simulated a phase separating membrane coupled to a picket-fence network resembling actin using a model similar to *Machta et al. (2011)* but extended to include membrane curvature. The energy contains three terms: $\mathcal{H}_{sim} = \mathcal{H}_{Helfrich} + \mathcal{H}_{Ising} + \mathcal{H}_x$ ('Materials and methods'). The first term describes the membrane elastic properties using the Helfrich form (*Helfrich, 1973*), with bending rigidity $\kappa$, and surface tension $\sigma$, the second term describes the phase separation using a conserved order parameter Ising model; the third term couples the phases to the local membrane curvature. The strength of the curvature coupling is proportional to the product of $\kappa$ and the spontaneous curvature difference between Lo and Ld domains ('Materials and methods'). Since there is a substantial range in the experimentally reported values of $\kappa$ and $\sigma$, a large uncertainty (about one order of magnitude) in the strength of the curvature coupling is implied (*Schick, 2012*). For this reason, we allow the curvature coupling strength to be scaled by a (dimensionless) factor $g$ in our analysis. The latter is defined such that, for $g > 0$, regions of positive curvature favor unsaturated lipids, that is Ld domains. For $g = 0$ our model reduces to the one of *Machta et al. (2011)*. In situations where curvature coupling is known to occur, one should restrict $g$ to finite positive values. The influence of the actin network is incorporated via (immobile) pinning sites, which are distributed randomly along the actin fibers (linear pinning density $\rho_p$). At the pinning sites, there is a preferred energetic attraction to one of the lipid species (set by the Lo%). The pinning sites also locally fix the membrane height $h$, which we model by keeping $h = 0$ at these locations (we assume the actin network to lie in a flat plane, providing the reference from which the membrane height is measured). Additionally, there is a steric repulsion between the membrane and the actin: directly underneath the actin fibers, the membrane height is restricted to negative values $h < 0$.

We first consider DPPE-biotin pinning sites that slightly prefer Lo domains (Lo% = 59 ± 5). Interestingly, the corresponding experiment (*Figure 3B*) revealed a positive PCC, implying a weak alignment of Ld domains along the actin fibers instead. This contradiction can be rationalized, however, when one considers the membrane curvature. In *Figure 4G*, we show how the simulated PCC varies with the curvature coupling parameter $g$, using pinning density $\rho_p = 0.1/nm$, corresponding to 20% of the total actin network being pinned. The key observation is that, at $g \approx 20$, the PCC changes sign and crosses the experimental value. We emphasize that the simulation data were not corrected for the optical point spread function (PSF) of the experiment; by artificially broadening the simulation images, lower values $g \sim 10$ were obtained, precluding a precise determination. In addition, the value of $g$, where the PCC changes sign also depends on the pinning density: by increasing $\rho_p$, also $g$ must increase to result in a positive PCC. In qualitative terms, the change of sign in the PCC reflects a competition between two effects: an energetic attraction between DPPE-biotin and saturated lipids, favoring alignment of Lo domains, vs a curvature-induced repulsion of these lipids away from the actin fibers. Due to the steric repulsion between the membrane and the actin fibers, the preferred curvature around the fibers is positive on average, thereby favoring Ld domains. At large $g$, the latter effect dominates, yielding a positive Pearson coefficient. In *Figure 4E*, we show a typical snapshot corresponding to $g = 20$ and $\rho_p = 0.1/nm$. In agreement with the experiment of *Figure 3B*, we observe a structure of small domains. In contrast, by using $g = 0$, the domains grow to be much larger (*Figure 4B*), contradicting the experimental observations.

Our model also predicts that, in the presence of curvature coupling, the pinning density required to induce domain alignment along the actin fibers can be much smaller. Previous simulations corresponding to $g = 0$ (*Ehrig et al., 2011*; *Machta et al., 2011*) required rather large pinning densities, $\rho_p \sim 0.2 - 1.25/nm$, in order to achieve this. In *Figure 4A*, we show a typical domain structure using DOPE-biotin pinning sites (Lo% = 11 ± 2) at pinning density $\rho_p = 0.1/nm$ in the absence of coupling to curvature ($g = 0$). *Figure 4D* shows the corresponding snapshot in the presence of curvature coupling,

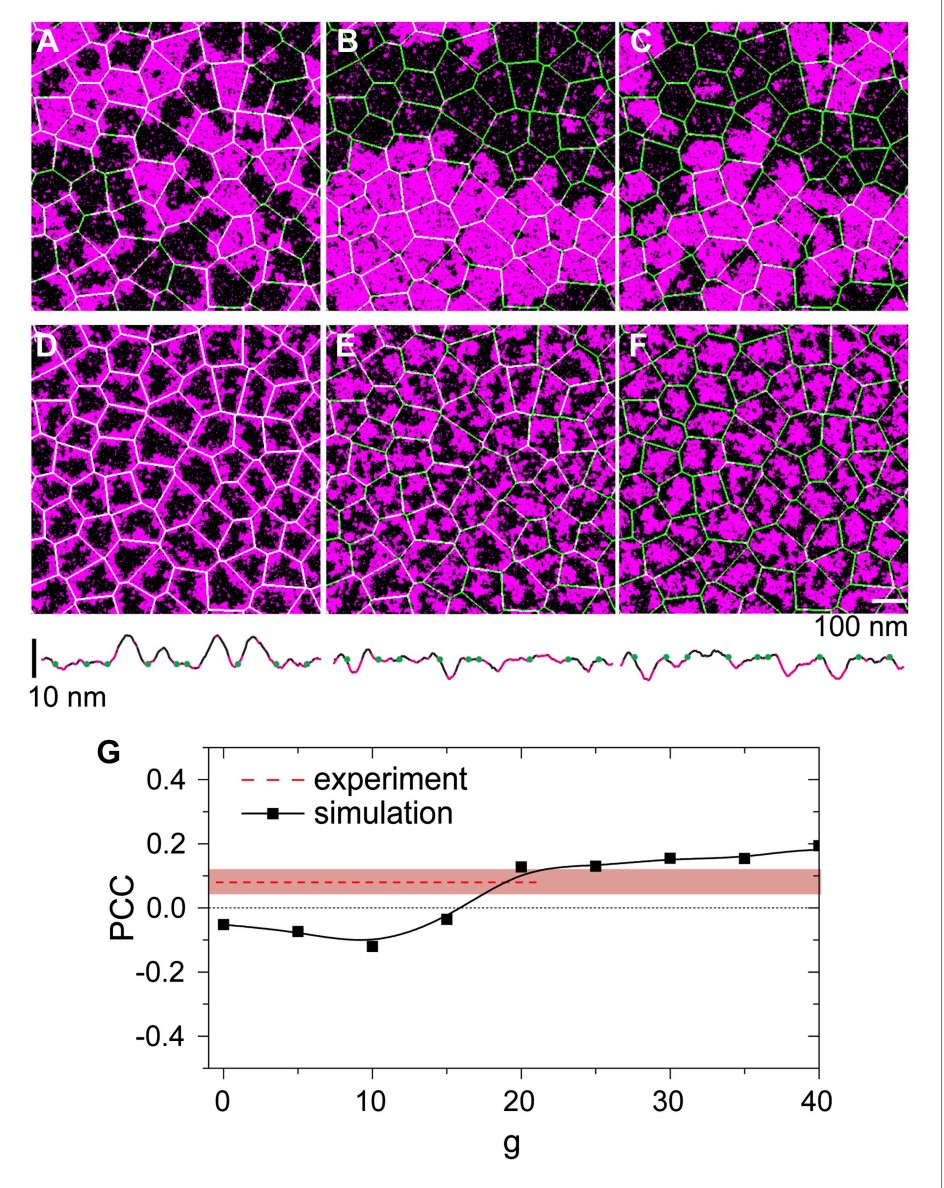

**Figure 4**. Simulation analysis of the influence of pinning and curvature coupling on lipid phase organization. All data refer to T = 19°C. The Ld lipids are shown as magenta, the Lo lipids as black, and the actin network is shown in green as an overlay. The pinning density $\rho_p$ = 0.1/nm. (**A–C**) Simulation snapshots obtained without coupling to curvature (g = 0) for the three species of pinning sites used in the experiments: Lo% = 11 ± 2 (**A**), Lo% = 59 ± 5 (**B**), and Lo% = 78 ± 7 (**C**). No significant influence of the actin network is apparent. (**D–F**) Same as (**A–C**) but in the presence of curvature coupling (g = 20). For snapshots (**D**) and (**F**), the lipid domains strongly correlate to the actin network, with Ld domains favoring actin in (**D**), and the inverse pattern in (**F**). The lower panels show height profiles of the images (**D–F**) scanned horizontally along the center of the image; the green dots indicate the positions of the actin fibers. (**G**) Pearson correlation coefficient PCC vs the curvature coupling g for the pinning species with Lo% = 59 ± 5. For weak curvature coupling, the PCC is negative indicating alignment of Lo domains along actin. By increasing the curvature coupling, the PCC becomes positive and 'meets' the experimentally observed value (conform *Figure 4B*).

using g = 20, which is the value of *Figure 4G* where the PCC changed sign. To reproduce the experimentally observed domain structure (*Figure 3A*) at low pinning densities, curvature coupling is thus essential. In *Figure 4D*, the average Ld domain size $R_{sim}$ ≈ 40 nm, which is somewhat below $R_{exp}$ ≈ 90 nm of the experiment (*Figure 1D*). Note, however, that the experimental value likely presents an

upper-bound, due to broadening by the PSF. The case of DSPE-PEG-biotin pinning sites (Lo% = 78 ± 7) is considered in *Figure 4C,F*. Again, curvature coupling is required to reproduce the (now inverted) domain structure of the experiment (*Figure 3C*). In the presence of curvature coupling, the length scale over which the effect of a pinning site propagates is thus enhanced dramatically. This is due to the elastic properties of the membrane: the bending rigidity and the surface tension define a length $\xi_h = (\kappa/\sigma)^{1/2}$ (*Schick, 2012*), which sets the scale over which the membrane height deformations propagate (for our model parameters $\xi_h \sim 100$ nm). When $g > 0$, this scale couples to the composition, in which case a reduced pinning density already suffices to induce domain alignment. We emphasize that $\xi_h$ is essentially independent of temperature, and so it is not necessary for the membrane to be close to a critical point.

Also included in *Figure 4* are typical membrane height profiles for the snapshots with curvature coupling (D,E,F) scanned along a horizontal line through the center of each image. These profiles qualitatively illustrate the curvature coupling effect: Ld domains (magenta) reveal positive curvature on average, while for Lo domains (black) the curvature is on average negative. In the absence of coupling to actin and $g = 0$, the typical root-mean-square height fluctuation is $h \approx 3.6$ nm. In the presence of actin and curvature coupling, these fluctuations are significantly reduced to $h \approx 2$nm, which is very close to the value reported in *Speck et al. (2010)*.

## Discussion

We have presented an experimental model system in which the response of membrane organization to a bound actin network can be accurately probed using superresolution STED microscopy and FCS. The application of our model system to a single component liquid-disordered membrane shows that actin has only a minor influence on the lateral distribution and dynamics of lipids (*Figure 1—figure supplement 2*, as well as *Figure 2A,B*). In contrast, under the same conditions using a ternary membrane, the effects are dramatic. In particular, by binding actin using pinning sites that attract the Ld phase, earlier simulation predictions (*Ehrig et al., 2011*; *Fischer and Vink, 2011*; *Machta et al., 2011*) could finally be put to a stringent test. In agreement with these simulations, our experiments confirm the absence of macroscopic phase separation in the presence of actin. Additionally, our experiments revealed the alignment of Ld domains along the actin fibers, leading to a channel-like domain structure very similar to structures observed in simulations (*Ehrig et al., 2011*; *Fischer and Vink, 2011*; *Machta et al., 2011*). Finally, in agreement with the simulations of *Machta et al. (2011)*, the enhanced diffusion of unsaturated lipids along the Ld channels, and the hindered diffusion of these lipids in directions perpendicular, was confirmed. These findings demonstrate that relatively simple simulation models are capable to capture key essentials of lateral membrane organization and dynamics.

However, by using pinning sites that weakly attract the Lo phase, our experiments also uncover phenomena that cannot be explained using these simulation models. The paradox is that, for the latter type of pinning site, one still observes alignment of Ld domains along the actin fibers, albeit weak. This indicates that there must be additional mechanisms at play—beyond the level of the pinning-lipid energetic interaction—playing a role in the lateral organization of the membrane. As possible candidate for such a mechanism, we considered the local membrane curvature, and a coupling of the latter to the lipid composition (*Schick, 2012*). A simulation model that incorporates these ingredients is able to reproduce the experimentally observed alignment of Ld domains, even when the pinning sites themselves energetically favor the Lo phase. The physical explanation is that the actin network locally induces regions with non-zero average curvature. The coupling of the curvature to the composition then causes these regions to prefer certain types of lipids. Provided the coupling is strong enough, it can overcome the pinning-lipid energetic interaction, which is how we interpret the experimental result. An additional finding is that, in the presence of curvature coupling, the effect of a pinning site extends over a much greater distance (set by the elastic properties of the membrane). Hence, the pinning density can be much lower compared to models where such a coupling is absent. We emphasize once more that curvature coupling is not the only conceivable mechanism that could account for our experimental findings. However, a recent experimental study (*Kaizuka and Groves, 2010*) of membrane phase separation using intermembrane junctions did uncover a very pronounced curvature coupling, making this a likely candidate.

There are interesting implications of our results concerning the in vivo organization of the plasma membrane. Our experiments show that the type of lipid domain selected to be stabilized depends

sensitively on the properties of the pinning species. A similar phenomenon, albeit on a larger scale, was observed by crosslinking GM1 with cholera toxin B (*Hammond et al., 2005*; *Lingwood et al., 2008*). Since the stabilized phase is determined by the properties of the pinning species, cells could locally sort their membrane components in this way. Moreover, this sorting mechanism persists to physiological temperatures, that is above the temperature of phase separation. At the same time, the pinning sites would naturally prevent the plasma membrane from phase separating at low temperatures. In the presence of curvature coupling, these effects are enhanced. We note that the proposed curvature effect in our experiments was likely limited by the Mica support. However, since the energy cost of lipid extraction far exceeds that of membrane de-adhesion from the support (*Helm et al., 1991*; *Lipowsky and Seifert, 1991*), we still expect some effect. In free-standing membranes, or in cell membranes, the curvature-coupling is anticipated to be stronger. The recent findings of *Kaizuka and Groves (2010)* seem to support this view.

## Materials and methods

### Preparation of mica supported membranes

Mica (Muscovite, Pelco, Ted Pella, Inc., Redding, CA) was cleaved into thin layers (~10 µm) and glued (optical UV adhesive No. 88, Norland Products Inc., Cranbury, NJ) onto clean glass cover slides. Immediately before spin-coating the lipid solution, the MICA on top of the glass was cleaved again to yield a thin (~1 µm) and clean layer. Next, 30 µl of 2 g/l lipid solution in Methanol/Chloroform (1:1) were spin-coated (2000 rpm, for 30 s) on top of the MICA. To remove residual solvent, the cover slide was put under vacuum for 20 min. The supported lipid bilayer was hydrated with warm (50°C) buffer (150 mM NaCl Tris pH 7.5) for 10 min and then rinsed several times to remove excess membranes until a single clean bilayer remained on the surface. All lipids were purchased from Avanti Polar Lipids, Inc., AL USA. The Ld phase was stained with far-red fluorescent DPPE-KK114 (*Kolmakov et al., 2010*) or green fluorescent DPPE-OregonGreen-488 (Invitrogen, Darmstadt, Germany). The Lo phase was stained with DSPE-PEG(2000)-KK114 or DSPE-PEG(2000)-Cromeo-488 (*Honigmann et al., 2013*). For imaging experiments, the concentration of fluorescent lipids was ~0.1 mol%; for FCS experiments ~0.01 mol% was used.

### Actin binding to supported membranes

Supported lipid bilayers were doped with biotinylated lipids (1,2-dioleoyl-sn-glycero-3-phosphoethanolamine-N-(cap biotinyl) (DOPE-biotin), 1,2-dipalmitoyl-sn-glycero-3-phosphoethanolamine-N-(cap biotinyl) (DPPE-biotin), 1,2-distearoyl-sn-glycero-3-phosphoethanolamine-N-[biotinyl(polyethylene glycol)-2000] (DSPE-PEG-biotin), also purchased from Avanti) that were used to bind actin fibers to the membrane. The following procedure was performed at 37°C to keep the membrane in the one phase region: The bilayer was incubated with 200 µl of 0.1 g/l streptavidin for 10 min and then rinsed several times to remove unbound streptavidin. Next, the membrane was incubated with 200 µl of 1 µM biotinylated phalloidin (Sigma-Aldrich, Steinheim, Germany) for 10 min and then rinsed several times to remove unbound phalloidin. Pre-polymerized actin fibers (500 µl with 7 µg/ml actin; Cytoskeleton Inc., Denver, USA) were then incubated with the membrane for 20 min and then rinsed several times to remove unbound actin. In case actin fibers were imaged, the actin was stained with green fluorescent phalloidin (Cytoskeleton Inc.). The membrane bound actin network was stable for at least 24 hr. The density of the actin network was controlled by the amount of biotinylated lipids in the membrane (*Figure 1—figure supplement 2*).

### Simulation model

The local membrane height $h(x,y)$ is a function of the lateral coordinates $x$ and $y$, which are discretized on the sites of a $L{\times}L$ periodic lattice, $L = 400a$, with lattice constant $a = 2$ nm. The membrane elastic energy $\mathcal{H}_{\text{Helfrich}} = \sum a^2\left(\kappa(\nabla^2 h)^2 + \sigma(\nabla h)^2\right)/2$, with the sum over all lattice sites, and $\nabla$ the gradient operator (*Helfrich, 1973*). The first term is the bending energy; the second term reflects the cost of area deformations. We use typical values, $\kappa \sim 2.7 \times 10{-}^{19}$Nm and $\sigma \sim 2 \times 10{-}^{5}$N/m, at the same time emphasizing that there is a considerable spread in the reported values of these quantities (*Schick, 2012*). This holds especially true for $\sigma$, whose value near a support may well be different. This, in turn, implies a large spread in the coupling to curvature strength (*Schick, 2012*). As stated before, we adopt the approach keeping $\kappa$ and $\sigma$ fixed, while allowing the curvature coupling strength to vary. To describe phase separation, we introduce the local composition $s(x,y)$, which reflects the lipid composition at site $(x,y)$. Experiments indicate that phase separation in membranes (without actin) is compatible

with the universality class of the Ising model (*Magde et al., 1972*; *Honerkamp-Smith et al., 2009*). We therefore use a two-state description, $s(x,y) = \pm 1$, where the positive (negative) sign indicates that the site is occupied by a saturated (unsaturated) lipid, leading to $\mathcal{H}_{Ising} = -J\sum s(x,y)s(x',y')$, with the sum over all pairs of nearest-neighboring sites, and coupling constant $J > 0$. To match the phase transition temperature of the Ising model to the experiment (*Machta et al., 2011*), we choose $J = 0.44 k_B T_c$, with $T_c = (273 + 28)K$ the transition temperature of the membrane without actin, and $k_B$ the Boltzmann constant. It has also been shown experimentally that the membrane height and composition are coupled via the local curvature (*Baumgart et al., 2003*; *Parthasarathy et al., 2006*; *Yoon et al., 2006*; *Parthasarathy and Groves, 2007*; *Kaizuka and Groves, 2010*). This motivates the term $\mathcal{H}_x = g\kappa\,\delta C\,a^2\sum s\nabla^2 h$, with the sum over all lattice sites, $\delta C \sim 10^6 \mathrm{m}^{-1}$ the difference in the spontaneous curvature between Lo and Ld domains (*Leibler and Andelman, 1987*; *Liu et al., 2005*; *Schick, 2012*), and $g > 0$ the dimensionless parameter introduced previously to reflect the fact that the model parameters are not known very precisely.

To include actin, a network of line segments (line thickness $a$) was superimposed on the lattice, with a typical compartment size ~100 nm, close to the experimental value (*Figure 1—figure supplement 2F*). This network was the Voronoi tessellation of a random set of points (*Ehrig et al., 2011*; *Machta et al., 2011*). The network was fixed to the membrane via pinning sites, which were immobile, and distributed randomly along the fibers. The actin network and the pinning sites couple to both the composition and the membrane height. To realize the former, we replaced the composition variable at each pinning site by a fixed value $s(x,y) = Lo\%/50 - 1$, where Lo% is the partitioning fraction of the pinning lipid derived experimentally (*Figure 3—figure supplement 1*). To couple the pinning sites to the membrane height, we imposed $h(x,y) = 0$ at the pinning sites (*Speck and Vink, 2012*). Additionally, we included a steric repulsion between the membrane and the actin fibers: lattice sites underneath an actin fiber have their corresponding height variable restricted to negative values.

## Monte Carlo simulation procedure

The model $\mathcal{H}_{sim}$ was simulated using the Monte Carlo method. To compute the gradient and Laplace operators, standard finite-difference expressions were used. The simulations were performed at conserved order parameter, using equal numbers of saturated and unsaturated lipids. Two types of Monte Carlo move were used. The first was a Kawasaki move (*Newman and Barkema, 1999*), whereby two sites of different composition were chosen randomly, and then 'swapped'. This move was accepted conform the Metropolis probability, $P_{acc} = \min\left[1, e^{-\Delta\mathcal{H}_{sim}/k_B T}\right]$, with $\Delta\mathcal{H}_{sim}$ the energy difference, $k_B$ the Boltzmann constant, and $T$ the temperature. The second move was a height move, whereby a new height was proposed for a randomly selected site; this height was optimally selected from a Gaussian distribution, as explained in *Speck and Vink, 2012*. We emphasize that the moves were not applied to pinning sites. In addition, there is the steric repulsion constraint at sites that overlap with the actin network: for these sites, height moves proposing a positive value were rejected. Kawasaki and height moves were attempted with equal a priori probability, with production runs typically lasting $2\cdot10^6$ sweeps, prior of which the system was equilibrated for $4\cdot10^5$ sweeps (one sweep is defined as $L^2$ attempted moves).

## Temperature control of the membrane

The temperature of the membrane and the surrounding buffer was controlled by a water cooled Peltier heat and cooling stage which was mounted on the microscope (Warner Instruments, Hamden, CT, USA). The achievable temperature range with this configuration was between 7 and 45°C, with a precision of 0.3°C. The actual temperature directly over the membrane was measured by a small thermo-sensor (P605, Pt100, Dostmann electronic GmbH, Wertheim-Reicholzheim, Germany).

## Microscopy

All experiments were performed on a confocal custom-built STED microscope whose main features are depicted in *Figure 1—figure supplement 4*. The confocal unit of the STED nanoscope consisted of an excitation and detection beam path. Two fiber-coupled pulsed laser diode operating at λexc = 635 nm and λexc = 485 nm with a pulse length of 80 ps (LDH-P-635; PicoQuant, Berlin, Germany) were used for excitation of the green and far red fluorescence respectively. After leaving the fiber, the excitation beams were expanded and focused into the sample using an oil immersion objective (HCXPLAPO 100x, NA = 1.4, Leica Microsystems). The fluorescence emitted by the sample was

collected by the same objective lens and separated from the excitation light by a custom-designed dichroic mirror (AHF Analysentechnik, Tuebingen, Germany). In the following, the fluorescence was focused onto a multi-mode fiber splitter (Fiber Optic Network Technology, Surrey, Canada). The aperture of the fiber acted as a confocal pinhole of 0.78 of the diameter of the back-projected Airy disk. In addition, the fiber 50:50 split the fluorescence signal, which was then detected by two single-photon counting modules (APD, SPCM-AQR-13-FC, Perkin Elmer Optoelectronics, Fremont, CA). The detector signals were acquired by a single-photon-counting PC card (SPC 830, Becker&Hickl, Berlin, Germany). The confocal setup was extended by integrating a STED laser beam. A modelocked Titanium:Sapphire laser (Ti:Sa, MaiTai, Spectra-Physics, Mountain View, USA) acted as the STED laser emitting sub-picosecond pulses around $\lambda$STED = 780 nm with a repetition rate of 80 MHz. The pulses of the STED laser were stretched to 250–350 ps by dispersion in a SF6 glass rod of 50 cm length and a 120 m long polarization maintaining single-mode fiber (PMS, OZ Optics, Ontario, CA). After the fiber, the STED beam passed through a polymeric phase plate (RPC Photonics, Rochester, NY), which introduced a linear helical phase ramp $0 \leq \Phi \leq 2\pi$ across the beam diameter. This wavefront modification gave rise to the doughnut-shaped focal intensity distribution featuring a central intensity zero. The temporal synchronization of the excitation and STED pulses was achieved by triggering the pulses of the excitation laser using the trigger signal from an internal photodiode inside the STED laser and a home-built electronic delay unit, which allowed a manual adjustment of the delay with a temporal resolution of 25 ps. The circular polarization of the STED and excitation laser light in the focal plane was maintained by a combination of a $\lambda$/2 and $\lambda$/4 retardation plates in both beam paths (B Halle, Berlin, Germany). Integration of a fast scanning unit enabled rapid scanning of the excitation and STED beam across the sample plane. A digital galvanometric two mirror-scanning unit (Yanus digital scan head; TILL Photonics, Gräfeling, Germany) was used for this purpose. The combination of an achromatic scan lens and a tube lens in a 4f-configuration (f = 50 mm and f = 240 mm, Leica, Wetzlar, Germany) realized a stationary beam position in the back aperture of the objective, preventing peripheral darkening within the focal plane at large scan ranges, such as vignetting. The maximal frequency of the Yanus scanner depended on the scan amplitude and varied between 2 and 6 kHz for scan amplitudes up to 150 µm, respectively. The hardware and data acquisition was controlled by the software package ImSpector (http://www.imspector.de/).

## Image analysis

To analyze the temperature-driven phase separation of the membrane (*Figure 1C*), we calculated the cumulant $U_1 = \left[x^2\right]/\left[|x|\right]^2$, where the square brackets $\left[\cdot\right]$ denote an average over all pixel values $x$ in the image. The latter were normalized beforehand such that the mean $\left[x\right] = 0$ and the standard deviation $\left[x^2\right] - \left[x\right]^2 = 1$. The average domain size $R$ (*Figure 1D*) was extracted from the radial distribution function $g(r)$ of the lipid intensity image, which represents the intensity correlations between two points of distance $r$. We observed that $g(r)$ was largest at $r = 0$, and decayed for $r > 0$. As a measure of the domain size $R$, we used the criterion $g(R) = 0.5 \times g(0)$. The Pearson correlation coefficient (PCC) between the actin and the lipid channels (*Figure 1E*) was calculated as the covariance of both channels divided by the product of the standard deviations of both channels. For each temperature up to 5 images from different parts of the sample were analyzed. Vesicles on top of the supported bilayer (which were visible in the lipid channel as round bright structures) were excluded from the analysis.

## Scanning FCS and pair correlation analysis

For the analysis of *Figure 2*, circular orbits 0.5–1.2 µm in diameter were scanned, at scanning frequency 4 kHz. The scanning orbit was subdivided into 64 pixels. For each pixel $i$, the fluorescence intensity $F_i(t)$ was recorded as a function of time $t$ for a duration of 30–60 s. The correlation between two pixels, $i$ and $j$, was computed via the pair correlation function (PCF) $G_{ij}(\tau) = F_i(t)F_j(t+\tau)/F_i(t)F_j(t) - 1$, where $\langle\cdot\rangle$ denotes a time average. *Figure 2A,C* shows the autocorrelation ($i = j$) of each pixel along the orbit, whereas *Figure 2B,D* shows the correlation between pairs of pixels $i$ and $j$ separated by a rotation of 180°. The maximum of the PCF yields an estimate of how long the fluorescent probes on average need to diffuse from $i$ to $j$ (*Digman and Gratton, 2009*). To avoid crosstalk between the two excitation spots, the distance between pairs (i.e., the diameter of the scanning orbit) was at least twice the size of the observation spot. In case of free Brownian diffusion, the PCF is homogenous around the scan orbit. In case the diffusion is hindered by obstacles, the maximum of the PCF is shifted to longer times, and its amplitude is decreased.

## Acknowledgements

We thank Vladimir Belov and Gyuzel Mitronova (MPI Göttingen) for the synthesis of the fluorescent lipid analogous, and Marcus Müller (Institute of Theoretical Physics, University of Göttingen) for stimulating discussions.

## Additional information

### Funding

| Funder | Grant reference number | Author |
| --- | --- | --- |
| German Research Foundation | SFB-937 (Project A6) | Alf Honigmann, Sina Sadeghi, Christian Eggeling, Richard Vink |
| Emmy Noether Program | VI-483 | Richard Vink |

The funders had no role in study design, data collection and interpretation, or the decision to submit the work for publication.

### Author contributions

AH, RV, Conception and design, Acquisition of data, Analysis and interpretation of data, Drafting or revising the article; SS, Conception and design, Acquisition of data, Analysis and interpretation of data; JK, Analysis and interpretation of data, Drafting or revising the article; SWH, Conception and design, Drafting or revising the article; CE, Conception and design, Analysis and interpretation of data, Drafting or revising the article

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
