## [Decision Letter]

Thank you for sending your work entitled “A lipid bound actin meshwork organizes liquid phase separation in model membranes” for consideration at *eLife*. Your article has been favorably evaluated by Randy Schekman (Editor-in-Chief) and 3 reviewers.

The Editor-in-Chief and the reviewers discussed their comments before we reached this decision, and the Editor-in-Chief has assembled the following comments to help you prepare a revised submission.

In this contribution by Sadeghi et al., the authors improve the understanding of how cellular membrane interactions with the cytoskeleton might affect the distribution of lipids. The authors use a biotin/streptavidin coupling approach to attach actin filaments to model membranes that are supported by a mica sheet. Model membranes consist of a lipid mixture with the propensity to show liquid ordered/liquid disordered phase coexistence. The authors show clear evidence for alignment of domains along actin fibers that might provide a reasonable mechanism for the regulation of cellular lipid phase domain size and distribution. The authors use a couple of cutting edge techniques for arriving at their conclusions, including two-point time correlation analysis of fluorescence fluctuations and diffraction limit breaking optical imaging. The findings of the manuscript are impressive and important. Nevertheless, I have several comments, the consideration of which might further improve the contribution.

1) There are a couple potential experimental pitfalls that the authors should acknowledge.

1A) First of all, the solid mica support likely perturbs the lipid bilayer by affecting domain structure and phase transition temperatures. I believe that such effects have been demonstrated by the groups of S. Keller and L. Tamm.

1B) Second, actin may interact with the mica support through defects in the lipid bilayer. Such an interaction may lead to irreversible bilayer disruption after membrane binding. The authors mention that significant correlation between spatial distribution of actin as well as domain distribution was found at a temperature as high as 37 degrees, but unless I have overlooked it there does not appear to be a description of at what temperature bilayers with bound actin actually become homogeneous. I am therefore wondering if the bilayer is irreversibly disrupted through actin/mica support interactions, or if the bilayer remains intact and reversibly responds to temperature changes.

1C) Third, the method with which the authors produce the lipid bilayer (spin coating of lipid solution, drying, and extensive rinsing), appears somewhat harsh to me regarding the integrity of the remaining single mica supported bilayer. Have the authors confirmed that rinsing does not lead to bilayer damage?

2) The organization of the manuscript can be improved.

2A) The order of the figure presentation would benefit from reconsideration, especially regarding a more linear presentation. As it stands, the authors explain Figure 1, then jump to Figure 2, then back to Figure 1, and then describe 2C.

2B) The description of Figure 3 is especially confusing. Figure 3 has three panels, however the text just mentions an upper and a lower panel. The “lower panel” as referred to in the main text appears to describe two different things: the middle panel, as well as the actual lower panel of the figure. Figure 3 is discussed before Figure 3. Figure 3 is not mentioned in the Figure caption. I recommend that the authors group six panels in this figure and label them A-F. The lower panel in Figure 3 contains interesting spatial information about the time autocorrelation obtained at two different points. I found the contents of this panel insufficiently explained though: why is there a strong angular dependence of the correlation function (in the absence of any actin in this case)?

3) I have some questions about the curvature aspects of the simulation.

3A) The authors introduce the parameter g as a coupling parameter between local composition and local membrane curvature. The term Hx, described in the supplement, however, also contains the bending stiffness and the difference of spontaneous curvature among the two phases. I would like to point out that the product of bending stiffness and spontaneous curvature difference already is the coupling parameter connecting curvature and local composition. This interpretation is consistent with the existing literature on curvature/composition coupling in membranes. The parameter g thus seems to be a (somewhat arbitrary) amplification factor that allows to dial up the interaction strength to make experimental data consistent with the simulations. I am not sure what the physical justification for this amplification factor is? In the absence of providing a physical justification, I would be forced to interpret g as a fudge factor.

3B) I am surprised that the authors consider deformations of a solid supported membrane through an actin filament. Perhaps naïvely, I would think that the adhesion energy between membrane and solid support would be sufficient to prevent such deformation. I believe that it should be possible to perform order of magnitude estimates, using measured values of the adhesion energy of a lipid bilayer on mica, as well as the work required to pull two lipids (per streptavidin) out of the solid supported membrane. Would de-adhesion be favored over lipid extraction, considering the density of biotin on the actin filament?

4) On occasion, the error analysis could be improved. For example, the authors quote measured partition coefficients, however, without providing uncertainty of the parameters.

5) The authors use a “typical” value for membrane tension of 2 × 10^−5^ N/m. As it stands, however, this value is arbitrary and it is possible that the actual value of membrane tension on the solid support is quite different from the value quoted. One reason for this is the already mentioned adhesion energy: I believe that the existing literature values for mica/lipid membrane adhesion energies would predict a membrane tension response toward forced unbinding from the substrate that is a bit higher than the above quoted value.

6) Notwithstanding point 5 above, it would be helpful if the authors considered alternative explanations for the surprising alignment of Ld domains with the actin fibers even for biotinylated Lo phase preferring lipids. Perhaps electrostatic interactions contribute; perhaps engagement with actin via streptavidin of a biotinylated lipid may lead to increased local membrane disorder?

7) The model system is supposed to consist of a single lipid bilayer sandwiched between a mica support and an actin network (linked by biotin – streptavidin – biotin). However, the method used to generate the lipid bilayer (drying of a lipid solution on the mica, followed by hydration) inevitably produces a stack of multiple bilayers. The authors claim that they obtained a single clean bilayer by repeated rinsing with buffer. I wonder if all bilayers except for the mica-bound one can be cleanly removed? The authors should do everything they can to determine if they have a single layer, although having multiple bilayers would not be a serious problem – other than that the reader would like to know.

8) This manuscript in its current form is that it does not situate the work very well with respect to the field. While some recent modeling and experimental studies of similar phenomena are appropriately mentioned, some key experimental reports even more closely related to the findings of this manuscript are omitted. These are detailed below, and they should be cited and discussed directly in the manuscript:

Hammond, A.T., F.A. Heberle,T. Baumgart, D. Holowka, B. Baird, G.W. Feigenson. “Crosslinking a lipid raft component triggers liquid ordered-liquid disordered phase separation in model plasma membranes.” PNAS, 102(18): 6320-6325, 2005.

This early paper details the critical fact that protein binding to membrane surface modulates miscibility phase transitions.

Yoshihisa Kaizuka and Jay T Groves. “Bending-mediated superstructural organizations in phase-separated lipid membranes.” New Journal of Physics 12 (2010) 095001 (11pp)

This more recent paper presents experimental observations of protein-mediated phase separation and its coupling to membrane curvature. This appears to be a very similar phenomenon to what the authors are reporting and should be discussed.

[Editors' note: further clarifications were requested prior to acceptance, as described below.]

Thank you for resubmitting your work entitled “A lipid bound actin meshwork organizes liquid phase separation in model membranes” for further consideration at *eLife*. Your revised article has been favorably evaluated by the Editor-in-Chief and one of the original reviewers. The manuscript has been improved but there are some remaining issues that need to be addressed before acceptance, as outlined below:

The revised article is significantly improved regarding its clarity and nuanced discussion of the experimental results and potential pitfalls regarding the interpretation of experimental findings.

My only remaining concern is that it does not seem to make physical sense to have a curvature coupling enhancement factor “g” that is equal to zero, when the curvature coupling factor deltaC (i.e., the spontaneous curvature difference between different phases) has a finite (non-zero) value.

---

## [Author Response]

*1) There are a couple potential experimental pitfalls that the authors should acknowledge*.

*1A) First of all, the solid mica support likely perturbs the lipid bilayer by affecting domain structure and phase transition temperatures. I believe that such effects have been demonstrated by the groups of S. Keller and L. Tamm*.

Yes, the electrostatic and frictional interactions of the lower leaflet of supported membranes with its substrate have been shown to change the membrane properties in comparison to free standing membranes. In general diffusion of lipids is slowed down by surface interactions. Depending on the preparation conditions of the supported membrane (ionic strength, temperature) the main phase transition (So/Ld) temperature of the lower and upper bilayer leaflet can be different, leading to decoupled domain formation in the two leaflets [1].

Temperature dependent domain (Lo/Ld) formation in spin-coated ternary membranes on mica similar to our system was investigated for example by Jensen et al. [2]. They also reported that the mica support influences the properties of the bilayer. Domains typically do not coarsen (grow/fuse) to a fully coalesced state as in free standing membranes, instead domain sizes remain in the low micrometer range. In presence of physiological ionic strength buffer (150mM KCl) the coupling of the membrane to the mica is reduced and domains are larger and round compared to hydration in pure water. If a second membrane is present on top of the first bilayer, in the second membrane domains will appear already at higher temperatures and they will coalesce similar to free standing membranes. This indicates that the (Lo/Ld) phase transition temperature of mica supported membranes appears to be lowered by the surface interactions.

In our system we also observe these reported support induced effects including the behavior of a second membrane on top of the first. Unfortunately, the second membrane is not stable enough to be used in our study. Even mild washing can remove these membrane patches. In any case, by characterizing the temperature dependent domain formation of the mica supported membranes in absence of actin and comparing the same system after the addition of an actin network we found clear changes induced by actin pinning which are not seen in the actin free supported membranes. We therefore think that the general conclusions of our work are valid also for free standing membranes.

*1B) Second, actin may interact with the mica support through defects in the lipid bilayer. Such an interaction may lead to irreversible bilayer disruption after membrane binding. The authors mention that significant correlation between spatial distribution of actin as well as domain distribution was found at a temperature as high as 37 degrees, but unless I have overlooked it there does not appear to be a description of at what temperature bilayers with bound actin actually become homogeneous. I am therefore wondering if the bilayer is irreversibly disrupted through actin / mica support interactions, or if the bilayer remains intact and reversibly responds to temperature changes*.

This is an important concern. Indeed, we observed that any visible defects in the membrane cause unspecific binding of streptavidin/actin to some extent. We therefore took great care during membrane preparation to avoid membrane defects. We checked the integrity of the membrane by fluorescence imaging and diffusion measurements with FCS after every preparation and rinsing step. In cases where visible defects in the membrane were present we did not observe that the membrane at the defects sites showed comparable phase behavior as in case of actin pinning on an intact membrane, e.g. no stabilization of phases was found at the defect sites. Additional evidence that the binding of actin to the membrane is not occurring via defects are:

1. The density of the actin meshwork is dependent on the concentration of biotin binding sites in the membrane (Figure 1—figure supplement 2).

2. The diffusion of lipids is not significantly altered after actin binding to single component membrane. Showing that actin binding does not induce membrane defects (Figure 1—figure supplement 2).

3. The Lo/Ld pattern observed in presence of actin is not seen at visible defect sites.

4. The phase alignment can be reversed by changing the anchor type to a Lo preferring lipid (Figure 3).

Indeed, the actin stabilized phases could not be completely homogenized by increasing the temperature above Tc, because 38°C was the highest temperature at which we could reliably image with our temperature control system. (Even though the temperature controller goes up to 45°C there is too much focal drift in the system to record high-resolution images.) Figure 2 shows that the correlation between actin and Ld domains decreases linearly above Tc. We therefore can estimate the temperature where the membrane would appear completely homogenous on our microscope to be T̴ 45°C. This is in agreement with the simulations of Machta et al. [3] which show clear cross-correlation between actin and lipid domains at 1.05Tc this corresponds to ∼ 43°C in our system. Also, it is not expected that there is a clear temperature threshold where the membrane becomes homogenous in case of actin pinning, because this transition is smooth, e.g., contrast and size of the stabilized domains just decreases as the temperature increases (see Machta et al., Figure 3). Nevertheless our membranes respond in a reversible way to temperature in the sense that the same phase pattern forms as the membrane is cooled back to room temperature.

*1C) Third, the method with which the authors produce the lipid bilayer (spin coating of lipid solution, drying, and extensive rinsing), appears somewhat harsh to me regarding the integrity of the remaining single mica supported bilayer. Have the authors confirmed that rinsing does not lead to bilayer*
*damage?*

We found that our method produces a homogenous bilayer with a very low defect rate. After hydrating the spin coated membranes usually the whole surface is covered with at least one membrane and at some parts patches of 2-3 membranes. Similar to the method described in Jensen et al. [2]. The membrane distribution was directly observed on our microscope (membrane stacks are visible as patches of 2-fold, 3-fold increased intensity). Rinsing was done while imaging the membrane on the microscope to have a direct feedback of the membrane quality. Additionally, we checked the diffusion constant of fluorescent lipids in the bilayer after each preparation step. In case of a damaged membrane the lipid diffusion is expected to drop. The diffusion constant of the lipids in the fluid phase of membrane after hydration was 3.5µm²/s which indicates that the membrane was intact. Up to 1mol% biotin binding sites in the membrane we found no significant change in lipid diffusion after the preparation steps (binding of actin) see Figure 1—figure supplement 2, indicating that the membrane was not damaged even after actin binding.

*2) The organization of the manuscript can be improved*.

*2A) The order of the figure presentation would benefit from reconsideration, especially regarding a more linear presentation. As it stands, the authors explain*
Figure 1*, then jump to*
Figure 2*, then back to*
Figure 1*, and then describe 2C*.

We have combined Figure 1 with Figure 2 to improve the reading flow.

*2B) The description of*
Figure 3
*is especially confusing.*
Figure 3
*has three panels, however the text just mentions an upper and a lower panel. The “lower panel” as referred to in the main text appears to describe two different things: the middle panel, as well as the actual lower panel of the figure.*
Figure 3
*is discussed before*
Figure 3*.*
Figure 3
*is not mentioned in the Figure caption. I recommend that the authors group six panels in this figure and label them A-F. The lower panel in*
Figure 3
*contains interesting spatial information about the time autocorrelation obtained at two different points. I found the contents of this panel insufficiently explained though: why is there a strong angular dependence of the correlation function (in the absence of any actin*
*in this case)?*

We apologize for the confusing representation. We now changed the figure and included new data. Instead of the old data on single component DOPC membranes without actin we included new data showing single component DOPC membranes with actin. We labelled the respective panels A-F and changed the description accordingly. Now the direct comparison between a single component and ternary membrane in presence of actin clearly shows that directional dependence of lipid diffusion is only found in a ternary membrane where the actin locally stabilizes lipid domains.

*3) I have some questions about the curvature aspects of the simulation*.

*3A) The authors introduce the parameter g as a coupling parameter between local composition and local membrane curvature. The term Hx, described in the supplement, however, also contains the bending stiffness and the difference of spontaneous curvature among the two phases. I would like to point out that the product of bending stiffness and spontaneous curvature difference already is the coupling parameter connecting curvature and local composition. This interpretation is consistent with the existing literature on curvature/composition coupling in membranes. The parameter g thus seems to be a (somewhat arbitrary) amplification factor that allows to dial up the interaction strength to make experimental data consistent with the simulations. I am not sure what the physical justification for this amplification factor is? In the absence of providing a physical justification, I would be forced to interpret g as a fudge factor*.

The referee is correct: The product of bending stiffness and spontaneous curvature is the coupling constant. The factor g was introduced because the numerical value of the coupling constant is not known very precisely. Depending on the properties of the membrane (bending stiffness, surface tension) the coupling strength can vary over one order of magnitude. The latter was pointed out by Micheal Schick [4], see the Discussion in the paragraph starting with the sentence: “Additional support for the mechanism proposed here is…” of the latter reference. We therefore allowed the strength to vary in our analysis, rather than choosing a single (unavoidably arbitrary) value.

This point is now mentioned in the main text.

*3B) I am surprised that the authors consider deformations of a solid supported membrane through an actin filament. Perhaps naïvely, I would think that the adhesion energy between membrane and solid support would be sufficient to prevent such deformation. I believe that it should be possible to perform order of magnitude estimates, using measured values of the adhesion energy of a lipid bilayer on mica, as well as the work required to pull two lipids (per streptavidin) out of the solid supported membrane. Would de-adhesion be favored over lipid extraction, considering the density of biotin on the actin*
*filament?*

Numerical estimates of the adhesion energy between a membrane and a mica support are reported to be in the range W=0.01−1 mJ/m² [5]. The energy of lipid extraction is much greater, at least 150 mJ/m², see caption of Figure 4 in the paper by Helm et al. [6]. Hence, we conclude that de-adhesion is favored over lipid extraction. This point is now mentioned in the text, including the above two references.

*4) On occasion, the error analysis could be improved. For example, the authors quote measured partition coefficients, however, without providing uncertainty of the parameters*.

We included error bars and provide the standard deviations in the text for the partitioning values as well as the Pearson coefficients.

*5) The authors use a “typical” value for membrane tension of 2 × 10*^*−5*^
*N/m. As it stands, however, this value is arbitrary and it is possible that the actual value of membrane tension on the solid support is quite different from the value quoted. One reason for this is the already mentioned adhesion energy: I believe that the existing literature values for mica / lipid membrane adhesion energies would predict a membrane tension response toward forced unbinding from the substrate that is a bit higher than the above quoted value*.

The tension used in our study was taken from the paper by Micheal Schick [4]. The latter was concerned with free-standing membranes, and so the referee is correct in pointing out that the tension in a supported system could be different. We now mention this in the text.

Measurements of the tension in a supported membrane are, however, difficult because of the substrate-membrane adhesion interaction that then must be added to the Hamiltonian. It is not obvious how to disentangle the tension contribution from the adhesion one, since both have the same dimension (i.e. energy per area). The issue here is whether, in the computation of the adhesion interaction, one should integrate over the projected membrane area, or the actual area. In case the latter convention is chosen, the adhesion interaction also contains a squared-gradient term, which “adds” to the tension term. Hence, in the presence of the support, the prefactor of the squared-gradient term reflects the sum of tension and adhesion contributions.

Because of all these difficulties, we adopted a pragmatic approach by choosing fixed values of the bending rigidity and the tension taken from the literature. The uncertainty in the latter quantities is then accounted for by allowing the strength of the curvature coupling to vary (i.e., the parameter g discussed in point 3A above, and now also discussed in the paper).

*6) Notwithstanding point 5 above, it would be helpful if the authors considered alternative explanations for the surprising alignment of Ld domains with the actin fibers even for biotinylated Lo phase preferring lipids. Perhaps electrostatic interactions contribute; perhaps engagement with actin via streptavidin of a biotinylated lipid may lead to increased local*
*membrane disorder?*

We included the possibility of steric and electrostatic interactions of the proteins with the membrane as an alternative mechanism for the observed Ld preference in case of low affinity Lo pinning sites in the discussion.

*7) The model system is supposed to consist of a single lipid bilayer sandwiched between a mica support and an actin network (linked by biotin – streptavidin – biotin). However, the method used to generate the lipid bilayer (drying of a lipid solution on the mica, followed by hydration) inevitably produces a stack of multiple bilayers. The authors claim that they obtained a single clean bilayer by repeated rinsing with buffer. I wonder if all bilayers except for the mica-bound one can be cleanly removed? The authors should do everything they can to determine if they have a single layer, although having multiple bilayers would not be a serious problem – other than that the reader would like to know*.

Yes, spin coating lipid membranes can produce stacked bilayers. However, the number of membranes can be controlled by the concentration of lipids in the spin coating solution. We performed a concentration row and inspected the resulting membranes on our microscope to find the right concentration for our experiments. At a lipid concentration of 2g/l usually the whole mica surface is covered with a single membrane and occasionally patches of 2-3 membranes. At lower lipid concentrations patches with no membrane are present. At higher concentrations multilayers are formed. The patches at 2g/l can be removed by controlled rinsing with a pipette. We do this step directly on the microscope to have a feedback how the membrane responds to the rinsing.

*8) This manuscript in its current form is that it does not situate the work very well with respect to the field. While some recent modeling and experimental studies of similar phenomena are appropriately mentioned, some key experimental reports even more closely related to the findings of this manuscript are omitted. These are detailed below, and they should be cited and discussed directly*
*in the manuscript:*

*Hammond, A.T., F.A. Heberle,T. Baumgart, D. Holowka, B. Baird, G.W. Feigenson. “Crosslinking a lipid raft component triggers liquid ordered-liquid disordered phase separation in model plasma membranes.” PNAS, 102(18): 6320-6325, 2005*.

*This early paper details the critical fact that protein binding to membrane surface modulates miscibility phase transitions*.

*Yoshihisa Kaizuka and Jay T Groves. “Bending-mediated superstructural organizations in phase-separated lipid membranes.” New Journal of Physics 12 (2010) 095001*
*(11pp)*

*This more recent paper presents experimental observations of protein-mediated phase separation and its coupling to membrane curvature. This appears to be a very similar phenomenon to what the authors are reporting and should be discussed*.

We agree that these reports are relevant for this manuscript. Both are now cited and discussed in the paper. We are thankful for the referee for having pointed out these studies, as they are nicely consistent with our own findings.

[Editors' note: further clarifications were requested prior to acceptance, as described below.]

*My only remaining concern is that it does not seem to make physical sense to have a curvature coupling enhancement factor “g” that is equal to zero, when the curvature coupling factor deltaC (i.e. the spontaneous curvature difference between different phases) has a finite (non-zero) value*.

By choosing g=0 the curvature coupling is turned off. This is appealing for theoretical modeling, as it allows for a smooth extrapolation between the model of Machta (g=0), and models featuring curvature coupling (g>0). Of course, to compare to experiments where curvature coupling is known to occur, one should restrict g to positive finite values, as the referee points out. In case deltaC is precisely known for the specific membrane system g can be omitted. To make the readers aware of this point, we have added the following sentence to the first paragraph of the ‘Simulation Results’ section: “In situations where curvature coupling is known to occur, one should restrict g to finite positive values.”

References

[1] A. Charrier and F. Thibaudau, “Main Phase Transitions in Supported Lipid Single-Bilayer,” *Biophys. J.*, vol. 89, no. 2, pp. 1094–1101, Aug. 2005.

[2] M. H. Jensen, E. J. Morris, and A. C. Simonsen, “Domain shapes, coarsening, and random patterns in ternary membranes,” *Langmuir ACS J. Surf. Colloids*, vol. 23, no. 15, pp. 8135–8141, Jul. 2007.

[3] B. B. Machta, S. Papanikolaou, J. P. Sethna, and S. L. Veatch, “Minimal model of plasma membrane heterogeneity requires coupling cortical actin to criticality,” *Biophys. J.*, vol. 100, no. 7, pp. 1668–1677, Apr. 2011.

[4] M. Schick, “Membrane heterogeneity: Manifestation of a curvature-induced microemulsion,” *Phys. Rev. E*, vol. 85, p. 031902+, Mar. 2012.

[5] R. Lipowsky and U. Seifert, “Adhesion of Vesicles and Membranes,” *Mol. Cryst. Liq. Cryst.*, vol. 202, no. 1, pp. 17–25, 1991.

[6] C. A. Helm, W. Knoll, and J. N. Israelachvili, “Measurement of ligand-receptor interactions,” *Proc. Natl. Acad. Sci.*, vol. 88, no. 18, pp. 8169–8173, Sep. 1991.